# Towards the Identification and Characterization of Putative Adult Human Lens Epithelial Stem Cells

**DOI:** 10.3390/cells12232727

**Published:** 2023-11-29

**Authors:** Pandi Saranya, Madhu Shekhar, Aravind Haripriya, Veerappan Muthukkaruppan, Chidambaranathan Gowri Priya

**Affiliations:** 1Department of Immunology and Stem Cell Biology, Aravind Medical Research Foundation, Madurai 625020, India; saranyapandi4995@gmail.com (P.S.); muthu@aravind.org (V.M.); 2Department of Biotechnology, Aravind Medical Research Foundation—Affiliated to Alagappa University, Karaikudi 630003, India; 3Cataract and IOL Services, Aravind Eye Hospital and Post Graduate Institute of Ophthalmology, Madurai 625020, India; madhushekhardr@aravind.org; 4Intraocular Lens and Cataract Services, Aravind Eye Hospital, Chennai 600077, India; haripriya@aravind.org

**Keywords:** human lens epithelial stem cells, adult tissue-resident stem cells, central zone, sphere forming ability, cataract lens

## Abstract

The anterior lens epithelium has the ability to differentiate into lens fibres throughout its life. The present study aims to identify and functionally characterize the adult stem cells in the human lens epithelium. Whole mounts of lens epithelium from donor eyes (normal/cataract) were immunostained for SOX2, gap junction protein alpha 1 (GJA1), PAX6, α, β and γ-crystallins, followed by a confocal analysis. The functional property of adult stem cells was analysed by their sphere forming ability using cultured lens epithelial cells from different zones. Based on marker expression, the lens epithelium was divided into four zones: the central zone, characterized by a small population of PAX6^+^, GJA1^−^, β-crystallin^−^ and γ-crystallin^−^ cells; the germinative zone, characterized by PAX6^+^, GJA1^+^, β-crystallin^−^ and γ-crystallin^−^; the transitional zone, characterized by PAX6^+^, GJA1^+^, β-crystallin^+^ and γ-crystallin^−^; and the equatorial zone, characterized by PAX6^+/−^, GJA1^+^, β-crystallin^+^, and γ-crystallin^+^ cells. The putative lens epithelial stem cells identified as SOX2^+^ and GJA1 membrane expression negative cells were located only in the central zone (1.89 ± 0.84%). Compared to the other zones, a significant percentage of spheres were identified in the central zone (1.68 ± 1.04%), consistent with the location of the putative adult lens epithelial stem cells. In the cataractous lens, an absence of SOX2 expression and a significant reduction in sphere forming ability (0.33 ± 0.11%) were observed in the central zone. The above findings confirmed the presence of putative stem cells in the central zone of the adult human lens epithelium and indicated their probable association with cataract development.

## 1. Introduction

The crystalline lens is an avascular, transparent biconvex structure in the eye, composed of a capsule, anterior lens epithelium, cortex and nucleus. It is entirely derived from a head epidermal ectoderm. The lens epithelium has the ability to divide and differentiate into lens fibres throughout its life and can develop into a complete lens with oriented lens fibres in vitro through neural retinal inductive tissue interactions in mice [1]. The anterior lens epithelium in animals (rat, mouse and bovine) has been divided into different zones—the central, germinative, transitional and/or equatorial zones—based on the morphology of cells [2], proliferative potential [3], expression of markers [2] and label retaining cell property (LRCs) [4]. At the equatorial zone, the meridional rows mark the boundary of the epithelium and the onset of fibre cell differentiation [5]. By tracing pulse labelled cells, it was identified that proliferating lens epithelial cells differentiate and migrate to the meridional row [3].

Lens regeneration has been reported after a lentectomy through the transdifferentiation of dorsal iris pigment epithelial cells or the cornea [6] and through spontaneous regeneration from residual lens epithelial cells following the removal of lens contents in several mammals [7]. In human lenses, epithelial cells in the central region were identified to have regenerative potential [8]. However, very little information is available on the various zones in the adult human anterior lens epithelium.

Adult tissue-specific stem cells have been found in almost all tissues of the human body and are responsible for maintaining homeostasis throughout life. These stem cells remain quiescent and divide if needed. Functionally, adult stem cells are characterized based on their slow cycling, LRC property [4], clonal analysis [9] and sphere forming ability [10].

Stem cells have been reported in the anterior lens epithelium, but their location remains controversial. Previous studies suggest that stem cells in the lens epithelium are located in the central zone based on (i) the localization of LRCs exclusively in the central zone of the mouse anterior lens epithelium [4], (ii) the development of a complete lens from the mouse anterior lens epithelium, excluding the equatorial region [1], (iii) the in vivo regeneration of a lens-like structure in congenital cataract patients, when the integrity of the central region (the lens capsule and lens epithelium) is maintained during a minimally invasive cataract surgery [8] and (iv) the expression of ABCG2/SOX2 in the central zone of the anterior lens epithelium using capsulotomy samples from cataract patients [11]. Contradicting the above observations, Yamamoto et al. (2008) [12] proposed the presence of stem cells expressing p75NTR and proliferating cell markers (PCNA, Cyclin D1, Cyclin B1 and Cyclin A1) in the region immediately anterior to the germinative zone of the mouse anterior lens epithelium. Further, the side-population cells with low Hoechst fluorescence (both blue and red) were located around the germinative zone in the mouse anterior lens epithelium [13]. A recent review of adult ocular stem cells indicated the presence of dividing lens epithelium stem/progenitor cells only in the equatorial zone, with the ability to differentiate into lens fibres [14].

Therefore, the present study focused on identifying and functionally characterizing the stem cells for lens epithelial cells in normal as well as cataractous human lens. The adult human anterior lens epithelium was divided into four zones based on the expression of the differentiated cell marker—a gap junction protein alpha 1 (GJA1) and crystallins. Since there are no exclusive markers for adult stem cells, putative lens epithelial stem cells were identified by combining two markers: (i) the transcription factor SOX2 and (ii) a differentiated cell marker GJA1. The expression of SOX2 has been reported to activate the development of the lens vesicle from the lens pit, and the weak expression of SOX2 along with high SOX1 expression have been found to be essential for the growth of the lens epithelium and fibre differentiation [15]. SOX2-expressing adult stem cells in mice have been traced to originate from foetal SOX2-positive epithelial progenitors [16]. SOX2, when co-expressed with Oct 4, Klf4 and c-Myc, was reported to de-differentiate somatic cells to a pluripotent state [17]. Hence, SOX2 was used as a positive marker to identify lens epithelial stem cells. Since reports on human epidermal stem cells and limbal epithelial stem cells indicated that these adult stem cells do not communicate through GJA1 and the expression of GJA1 was associated with differentiation, GJA1 was included as a negative marker [9,18,19].

SOX2^+^ cells negative for membrane GJA1 expression were defined as putative lens epithelial stem cells and were identified to be located only in the central zone of the human anterior lens epithelium. The significantly higher percentage of sphere formation (including SOX2^+^, GJA1^−^ and PAX6^+^ cells) from this zone confirmed the location. In the cataractous lens epithelium, the absence of SOX2 expression in the central zone as well as a significant reduction in their sphere forming ability indicated the loss of these putative stem cells.

## 2. Materials and Methods

### 2.1. Sample Collection

The whole and excised globes (after the removal of the cornea for transplantation) from donors having phakic lenses were obtained from Rotary Aravind International Eye Bank, Aravind Eye Hospital, Madurai. The eyes were procured by the eye bank after obtaining consent from the donor’s family. The inclusion criteria for the selection of tissues were (i) eyes enucleated within 6 h of death and received at the research laboratory within 24 h, (ii) donor globes with normal and cataract lenses (opacification was observed during slit lamp examination at the eye bank) and (iii) donors with no history of ocular infection or systemic disease. Eyes from donors whose cause of death was due to either poison or a snake bite were excluded from the study. The human lenses from normal—31 donors, aged 40.65 ± 14.68 (mean ± SD) years (range: 20 to 71 years) and cataract—6 donors, aged 61.3 ± 13 (mean ± SD) years (range: 50 to 82 years) were included in the study. This study adhered to the tenets of the Declaration of Helsinki and was approved by the Institutional Review Board of the Aravind Medical Research Foundation and the Institutional Committee for Stem Cell Research (IRB2018015BAS).

### 2.2. Human Tissue Preparation

#### Whole Mount of the Human Anterior Lens Epithelium

The lens was positioned with the posterior side facing up under a stereo zoom microscope (Nikon SMZ645, Tokyo, Japan), and the posterior capsule was peeled [20]. The lens fibres were then gently removed. The whole mount of lens epithelium along with the lens capsule was used for immunostaining (Figure 1).

**Figure 1 cells-12-02727-f001:**
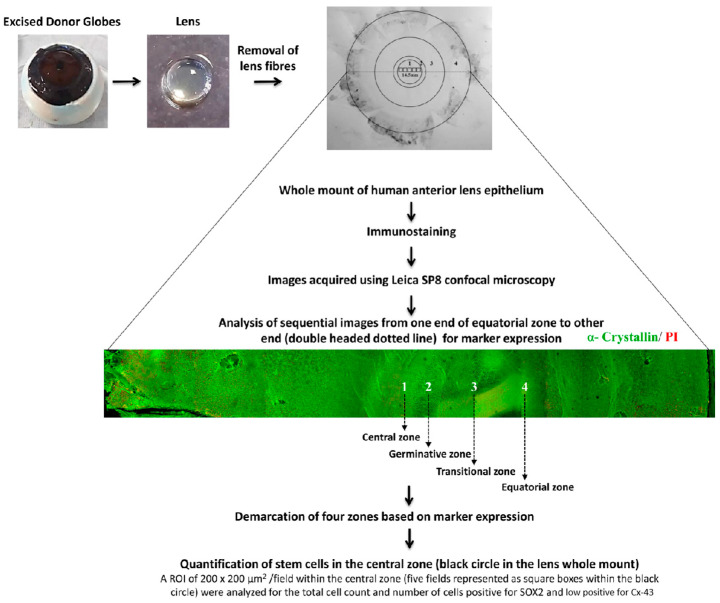
Characterization of human anterior lens epithelium: The human lens was dissected from the excised donor globe. After removal of the lens fibres, the whole mount of the human anterior lens epithelium was immunostained for various markers, and images were acquired sequentially from one end of the whole mount to the other end using Leica SP8 confocal microscopy. A representative montaged confocal image of the human anterior lens epithelial whole mount immunostained for α–crystallin (FITC-green) and counterstained with propidium iodide (PI-red) is given. The four zones of the anterior lens epithelium (1-central zone; 2-germinative zone; 3-transitional zone; 4-equatorial zone) were identified based on the marker expression (Table 1). Quantification of the stem cells in the central zone (black circle) was carried out in five consecutive confocal images (represented as small black squares) using Leica software LAS AF 3.3.0.10134 (*n* = 3 donors). Within an ROI of 200 × 200 μm^2^/image, the total number of lens epithelial cells (based on nuclear stain) and the number of SOX2^+^ GJA1^−^ cells were counted.

**Table 1 cells-12-02727-t001:** Markers to demarcate the different zones in the human anterior lens epithelium. On the basis of the literature and the present study, the human anterior lens epithelium was identified and demarcated into four zones based on marker expression. The symbols “+”, “−”, and “+/−” stand for the presence of cells expressing a marker, the absence of marker expression, and both respectively.

Zones	Marker Expression									
	Based on the Literature					Present Study				
	SOX2 [16]	GJA1 [21]	β-cry [2]	γ-cry [2]	α-cry [22]	SOX2	GJA1	β-cry	γ-cry	α-cry
Central zone	−	+	−	−	+	+/−	−/+	−	−	+
Germinative zone	−	+	−	−	+	−	+	−	−	+
Transitional zone	−	+	+	−	+	−	+	+	−	+
Equatorial zone	+	+	+	+	+	−	+	+	+	+

### 2.3. Immunostaining

The entire anterior lens epithelium was spread on a glass slide, fixed either in 4% paraformaldehyde (PFA) (P6148, Sigma-Aldrich Corp., St. Louis, MO, USA) or methanol for 15 min and blocked with the avidin biotin blocking system. After overnight incubation at 22 °C with the primary antibody, the corresponding biotinylated secondary antibody was added and incubated for 1 h at 22 °C. Streptavidin fluorescein isothiocyanate (FITC, BD Pharmingen—San Diego, CA, USA) was added at a dilution of 1:1000 (in 1X PBS) and incubated for 1 h at 22 °C in the dark to visualize the marker. For double immunostaining, the anterior lens epithelium was incubated overnight at 22 °C with anti-connexin 43 (GJA1) antibody (the second primary antibody), followed by the corresponding biotinylated secondary antibody and streptavidin-Alexa 633 (Thermofisher Scientific—Waltham, MA, USA) at a dilution of 1:1000 in 1X PBS or with phalloidin—TRITC (1:400). Between steps, the slides were washed three times in 1X PBS, and the stained whole mounts were mounted with Vectashield mounting medium containing 4,6-diamidino-2-phenylindole (DAPI)/propidium iodide (PI) (H-1200-10/H-1300-10, Burlingame, CA, USA). During the immunostaining, the whole mount without the addition of primary antibody was used as a negative control. Primary and secondary antibodies used are listed in Table 2.

### 2.4. Confocal Microscopy Imaging

Acquisition of the immunostained whole mounts was carried out using a laser scanning microscope (Leica SP8 confocal microscope, Wetzlar, Germany), as previously described (Sunderasan et al., 2019) [23]. Briefly, fluorescent z-stack images were acquired using the following settings: the emission band width for DAPI ranged from 358 to 441 nm and FITC from 496 to 535 nm using laser blue 488; for PI/TRITC, from 555 to 600 nm using laser green 552, and for Alexa Fluor 633 from 640 to 725 nm using laser red 633. Using the above-mentioned parameters, the images were acquired horizontally from one end to the other end of the whole mount of the anterior lens epithelium (Figure 1) using 10× as well as 63× objective.

### 2.5. Image Analysis

Sequential images were analysed from one end to the other end and montaged to represent the whole mount of the anterior lens epithelium (Figure 1). The total diameter of the whole anterior lens epithelium was measured horizontally using Leica Software (LAS AF 3.3.0.10134). The demarcation of four zones was carried out based on the measurement of regions positive/negative for GJA1, β-crystallin and γ-crystallin. The plasma membrane expression of GJA1 was defined by analysing the z-stack images of the positive cells (Appendix A). The putative lens epithelial stem cells were identified as cells expressing the stem cell marker/transcription factor SOX2 but negative for the membrane expression of the differentiated cell marker GJA1. For each donor tissue (*n* = 3 donors), five fields from the central zone were analysed for putative stem cell content. The percentage of putative lens epithelial stem cells was calculated by counting the total number of epithelial cells and the total number of putative stem cells in an ROI of 200 × 200 µm^2^/field. The results were represented as the mean percentage ± standard error mean (SEM).

### 2.6. Lens Explant Cultures

The human anterior lens epithelial explant cultures were established from normal (*n* = 5) and cataractous (*n* = 3) human anterior lens epithelium as described by Lin et al., (2016) [8]. The whole mount of lens epithelium was prepared, and the explants were cut into 1 × 1 mm pieces from the central, germinative + transitional (since the germinative zone was too small (0.2 mm) to be separated, it was dissected with transitional zone) and equatorial zone. Each explant was allowed to attach to a 35 mm dish (153066, Thermo Fisher Scientific, Waltham, MA, USA) coated with Matrigel (1:100) (354230, Corning, Tewksbury, MA, USA). The explants were then incubated at 37 °C for 4–6 min, and 1 mL of lens epithelial cell (LEC) medium (MEM (11095080, Gibco^TM^, Life Technologies, Grand Island, NY, USA), 20% foetal bovine serum (10270106, Gibco^TM^, Life Technologies, Paisley, UK), non-essential amino acid (M7145, Sigma-Aldrich, Gillingham, UK), penstrep (15140122, GibcoTM, Life Technologies, NY, USA), gentamicin (15710064, Gibco^TM^, Life Technologies, NY, USA), SB431542 (04001010, Stemgent-REPROCELL, Beltsville, MD, USA)) was added. The growth was monitored under an inverted phase contrast microscope (Nikon Eclipse TS100, Tokyo, Japan), and the medium was changed on alternate days for two weeks.

### 2.7. Establishment of Lens Sphere Cultures

As described by Sundaresan et al. (2021) [24], the cultured cells were trypsinized, and 1 × 10^3^ viable cells were seeded on ultra low attachment dishes (3471, Corning, Kennebunk, ME, USA). Further, the cells were maintained in sphere medium containing DMEM/F12 (1:1) + Glutamax^TM^ (10565018, Gibco^TM^, Life Technologies, NY, USA), 2% B27 (17504044, Gibco^TM^, Life Technologies, NY, USA), 20 ng/mL epidermal growth factor (PHG0311, Thermo Fisher Scientific, MA, USA), 20 ng/mL fibroblast growth factor (13256029, Gibco^TM^, Thermo Fisher Scientific, MA, USA), 5 μg/mL heparin (375095, EMD Millipore Corp., Danvers, MA, USA), 0.1 mg/mL bovine serum albumin (B8894, Sigma), 2.5 mM L-glutamine (35050061, Gibco^TM^, Life Technologies, NY, USA), and 250 μg/mL penicillin-streptomycin (15140122, GibcoTM, Life Technologies, NY, USA), and cultured. After 7 days of culture, spheres were imaged using an inverted phase contrast microscope (Nikon Eclipse TS-100, Tokyo, Japan) fitted with a camera, and ImageJ (https://imagej.nih.gov/ij/) (accessed on 21 November 2023) was used to determine the diameter of the spheres. For quantification, the total number of spheres ≥ 60 μm in diameter was counted in each dish, the percentage of spheres formed per 10^3^ cells seeded was calculated, and the data were represented as mean ± SD. The statistical significance of the sphere forming ability of cultured cells from different zones was analysed by the Kruskal–Wallis test and between normal and cataract lens epithelial cells by the Wilcoxon rank sum test. Further, the lens spheres were centrifuged at 400 rpm for 3 min using a cytospin system (Thermo Shandon—Pittsburgh, PA, USA). The lens sphere cytosmears were fixed with 4% PFA, followed by immunostaining for SOX2, GJA1, and PAX6 as described under Section 2.3.

## 3. Results

### 3.1. Identification of Four Zones in Human Anterior Lens Epithelium

The demarcation of each zone was based on the expression of specific markers (Table 1). Analysis of the sequential confocal images of the immunostained lens epithelial whole mounts from one end of the equatorial zone to the other (*n* = 3 donor eyes/marker) identified that the gap junction protein GJA1 was expressed by all epithelial cells in the germinative, transition, and equatorial zones and only in 90.83 ± 5.9% of cells in the central zone (Figure 2B). The early elongation marker β-crystallin was positive in all cells of the transition and equatorial zones but was totally negative in the central and germinative zones (Figure 2C). The expression of the late elongation cell marker γ-crystallin was observed in all cells of the equatorial zone, while cells in the other zones were negative (Figure 2D). The neuroectodermal marker PAX6 was strongly expressed by almost all epithelial cells in the central, germinative and transitional zones and by 10.09 ± 0.12% of cells in the equatorial zone (Figure 2E). The lens epithelial marker α-crystallin was expressed by all of the epithelial cells in the four zones (Figure 2F).

The montaged images of the anterior lens epithelial whole mounts showed that the total length of the epithelium was 14.5 ± 0.7 mm (Figure 3). The central zone, measuring 1.66 ± 0.1 mm, included a population of GJA1^−^, β-crystallin^−^, γ-crystallin^−^, and α-crystallin^+^ cells. The germinative zone, consisting of GJA1^+^, β-crystallin^−^ and γ-crystallin^−^ cells, encompassed a region 0.2 mm peripheral to the central zone. The transitional zone with GJA1^+^, β-crystallin^+^, and γ-crystallin^−^ cells was identified to be 2.17 mm peripheral to the germinative zone. The equatorial zone, measuring 4.05 mm from the transitional zone, was characterized by GJA1^+^, β-crystallin^+^ and γ-crystallin^+^ cells.

### 3.2. Identification and Location of Putative Adult Stem Cells in the Human Anterior Lens Epithelium

The putative adult stem cells in human lens epithelium were identified by the expression of the transcription factor SOX2. For confirmation, double immunostaining with the differentiated epithelial cell marker GJA1 was carried out (Figure 4A). The cells that expressed SOX2 and were negative for GJA1 membrane expression were defined as putative lens epithelial stem cells (Figure 4B). The sequential z-stack images of Figure 4B are given in Appendix A. Immunostaining for SOX2 and phalloidin highlighted the outline and structure of the lens epithelial cells (Figure 4C). Further, the sequential confocal images identified SOX2 expression in 9.16 ± 10.2% of cells in the central zone, while cells in the germinative, transition and equatorial zones were all negative for this marker (Figure 2A). The presence of SOX2^+^ GJA1^−^ cells only in the central zone indicated that the putative stem cells for the human anterior lens epithelium are located in this region and not in other zones of the human anterior lens epithelium (Figure 4A).

Double immunostaining for SOX2 and GJA1 identified three populations of cells in the central zone (Figure 4B): (i) SOX2^+^ GJA1^−^ putative stem cells, (ii) SOX2^+^ GJA1^+^ cells, and (iii) SOX2^−^ GJA1^+^ cells. Further quantification of cells in an ROI of 200 × 200 µm^2^ from five fields in the central zone per whole mount (*n* = 3) using ImageJ identified 1.89 ± 0.84% cells to be SOX2^+^ GJA1^−^ cells, 7.27 ± 5.07% to be SOX2^+^ GJA1^+^ cells, and the remaining 90.83 ± 5.9% to be SOX2^−^ GJA1^+^ cells (Table 3). All of the epithelial cells in the germinative, transition, and equatorial zones were SOX2^−^ GJA1^+^. Thus, the putative lens epithelial stem cells identified as SOX2^+^ GJA1^−^ cells were confined to the central zone of the human lens epithelium.

The nuclear expression of SOX2 was absent in all of the cataract lens epithelium analysed (Figure 5A). Further, the presence of autofluorescing vacuoles (UV to far red range) was identified in a subset of cells only in the central zone of these cataractous donor lenses (Figure 5B).

### 3.3. Functional Characterization of Lens Epithelial Cells—Sphere Forming Ability

The sphere forming ability of anterior lens epithelial cells was evaluated using explant cultures established from the central, germinative + transitional and equatorial zones. In a normal lens, cells in the outgrowth from the central zone consisted of cuboidal epithelial cells, while germinative and transitional zones showed a mixed population of epithelial and fibre-like cells, and the equatorial zone revealed the migrating cells from the explant to be fibre-like cells. In the cataractous lens, the outgrowth from the central zone as well as germinative+transitional zones were a mixed population of epithelial and fibre-like cells, while those from the equatorial zone were fibre-like cells (Figure 6).

After 7 days of culture, the cells from the central zone were identified to have a significantly higher percentage of spheres-1.68 ± 1.0% (*p* = 0.007) compared to other zones (germinative + transitional zones-0.20 ± 0.28%and equatorial zone-0.04 ± 0.08%, Figure 7A). A significant reduction in the percentage of sphere forming ability (0.33 ± 0.11%) (*p* = 0.025) was observed in the cultured central zone cells from the cataractous lens epithelium, and no spheres were observed with the cells from other zones (Table 4).

Characterization of lens spheres (a minimum of three spheres from each culture) for the expression of the stem cell marker SOX2 revealed that the normal lens epithelium-derived spheres had SOX2 expression in almost all cells, whereas cataract lens epithelium-derived spheres had one or two cells with low expression (Figure 7B). Double immunostaining of spheres derived from normal central zone lens epithelial cells revealed the presence of SOX2 positive but GJA1 negative cells, confirming that the putative adult lens epithelial stem cells were localised in the central zone (Figure 7C). Further, the expression of PAX6 by all cells in the sphere confirmed the neuroectodermal origin of these lens epithelial cells (Figure 7B).

## 4. Discussion

Adult tissue-resident stem cells are quiescent but have the ability to self-renew and to differentiate in response to the need, to maintain the tissue homeostasis throughout life [25]. These stem cells constitute 0.1–10% of cells in each tissue. They are mostly unipotent, lineage-specific and maintained in a unique microenvironment or niche. Adult tissue-resident stem cells are defined based on their phenotypic and functional characteristics. While there are no exclusive markers for adult stem cells, a combination of two or more markers/parameters is essential to identify stem cells [23,26,27,28]. Functionally, adult stem cells are defined based on their label-retaining cell property [2,24,29], clonal ability [9,30,31], sphere forming ability [10,24,32], side population property [33,34,35], ability to regenerate lineage-specific cells [8,36,37] and lineage tracing experiments [16,38,39]. With ageing, the adult stem cells decrease in number, as shown in human trabecular meshwork and dental pulp [23,40]. At the same time, ageing is also associated with a decrease in their function in some tissues, such as hematopoietic stem cells and hair follicle stem cells [41,42]. Therefore, understanding the basic biology of these adult tissue-resident stem cells is still evolving, as is their role in homeostasis, during ageing as well as in age-related diseases. In this study, the putative adult stem cells in the human anterior lens epithelium were characterized based on the immunostaining for stem cell marker SOX2 along with differentiated cell marker GJA1 (plasma membrane expression) and by sphere forming assay in normal and cataractous lens. 

Characterization of the anterior lens epithelium has been carried out mostly in animals at early stages of development [3,22,43,44] based on their proliferative potential: the central zone with a low proliferative index, the germinative zone with a high proliferative index, and no proliferation in the transitional zone [45]. Ong et al. (2003) demonstrated different zones based on the cellular morphology (central, germinative and transitional zone) and indicated that the expression of β-crystallin, an early elongation marker (medial transition zone), preceded γ-crystallin, a late elongation marker (superficial cortical fibre region), in the bovine lens [2]. Based on this, in the present study, the β-crystallin^+^ γ-crystallin^−^ cells identified the transitional zone, and β-crystallin^+^ γ- crystallin^+^ cells the equatorial zone. While α-crystallin was expressed by all of the lens epithelial cells as reported earlier [22], a small population of cells in the central zone were negative for the membrane expression of GJA1, the differentiated epithelial cell marker. All cells in the germinative zone were GJA1^+^, β-crystallin^−^ and γ-crystallin^−^, distinguishing them from the central zone. The expression of PAX6 in the central to transitional zones of the human lens epithelium highlighted the neuroectodermal origin. Absence of PAX6 expression in γ-crystallin positive cells in the equatorial zone confirmed the previous report that the expression of PAX6 mRNA was lower in lens fibres compared to lens epithelium [46]. Since there are no previous studies on adult human lens epithelium, additional studies on the proliferative status of the cells are essential to confirm this demarcation of zones as reported in animal studies.

The putative adult tissue-resident stem cells were identified by immunophenotyping using the transcription factor SOX2 as a marker. SOX2 has been reported as a stem cell marker in several tissues, including human limbal epithelial [47], Muller glial [48], and neural stem cells [49], cultured human foetal lens epithelial cells [8], and several epithelial tissues of adult mice [16]. Kamatchi et al. (1998) demonstrated, in chick embryo, the sequential expression of SOX genes: strong expression of SOX2 at the stage of lens pit, lens vesicle, and lens epithelial formation [15] and replacement of SOX2 by SOX1 at the stage of lens fibre development [50]. The expression of SOX1 was significant for γ crystallin gene expression and elongation of lens fibre cells. Thus, the SOX genes—SOX2 and SOX1—are expressed during lens epithelium and fibre differentiation, respectively. Lineage tracing experiments in mice confirmed that the SOX2^+^ adult stem cells, including the lens, originate from foetal SOX2^+^ epithelial progenitors during development. The SOX2^+^ cells were defined as stem cells, and SOX2^−^ cells as differentiated cells [16]. The expression of SOX2 in human lens epithelium has been reported earlier by Fu et al. (2016), wherein they identified the expression of stem cell markers SOX2, ABCG2 and Ki67 in the central zone of younger people aged 0–10 years. Lin et al. (2016) demonstrated the presence of SOX2^+^, PAX6^+^, and BMI1^+^ stem cells in cultured human lens epithelial cells. In this study, sequential analysis of the native human anterior lens epithelium (horizontally from one end of the equatorial zone towards the other end) from donors <60 years of age identified the presence of cells expressing the stem cell marker SOX2 only in the central zone, accounting for 9.16 ± 5.89% of central zone cells, similarly to the observations of Fu et al. (2016) in young donors. 

In addition to using the positive expression of SOX2, we also used a negative marker to identify the adult lens epithelial stem cells. The significance of combining two parameters to identify stem cells was established in our previous studies on limbal epithelial stem cells [24], buccal mucosal epithelial stem cells [51], and trabecular meshwork stem cells [23], as well as by other groups [52,53]. In our study, the expression of SOX2 was analysed along with the membrane expression of the differentiated epithelial cell marker GJA1, which is one among the 10 connexins reported to be expressed in the differentiated cells in the human eye. Reports are available on the application of GJA1 as a negative marker in human limbal epithelial stem cells [19,26,54] and adult epidermal stem cells [18]. Absence of intercellular communication through the gap junction, specifically GJA1, might reflect the ability of the stem cells to retain the stemness in their microenvironment [19]. The plasma membrane expression of GJA1 was analysed in the present study using sequential confocal z-stack images. Double immunostaining with a plasma membrane marker is essential to confirm the membrane expression of GJA1 in the human lens epithelium.

Among the SOX2^+^ cells identified in this study in the central zone, a fraction (1.89 ± 0.84%) was negative for the differentiated cell marker GJA1 (plasma membrane expression). Based on previous studies on GJA1 expression in adult ocular stem cells [19,26,54] and other stem cells [18,51], we hypothesize that the SOX2^+^ GJA1^+^ cells are derived from SOX2^+^ GJA1^−^ stem cells. Hence, we define cells positive for SOX2, PAX6, and α-crystallin and negative for GJA1 membrane expression, β and γ crystallins as putative lens epithelial stem cells. Further molecular studies are essential to elucidate the difference between SOX2^+^ GJA1^−^ and SOX2^+^ GJA1^+^ cells and to confirm that the SOX2^+^ GJA1^−^ cells represent stem cells as well as to identify additional markers for the adult stem cells in the human lens epithelium. 

For confirming the location of stem cells, the most widely used functional assay, the sphere forming assay, was carried out using cultured human lens epithelial cells from different zones. The formation of spheres under low adherence culture conditions has been reported in other tissues of the human eye, such as trabecular meshwork stem cells [24] and retinal pigment epithelial stem cells [55]. In the present study, the formation of a higher percentage of spheres by the cultured cells from the central zone (with cobblestone morphology) was demonstrated. In addition, most of the cells in these spheres were positive for SOX2 but negative for plasma membrane GJA1 expression and PAX6 positive, indicating the presence of a subpopulation of SOX2^+^ cells that represent stem cells. 

In this study, we defined the putative stem cells based on the presence of a small population of SOX2^+^ cells that were negative for membrane expression of GJA1 in the central zone of native human lens epithelium and the presence of such cells (SOX2^+^, GJA1^−^, PAX6^+^) in the spheres derived from cultured central zone cells. Further, the phenotypic (SOX2^+^, PAX6^+^, GJA1^−^, β crystallin^−^, γ crystallin^−^ cells) and functional characterization (sphere formation) confirmed the putative stem cells to be located in the central zone of the human anterior lens epithelium. Having identified the location of putative stem cells in normal human anterior lens epithelium, the loss of SOX2^+^ cells in the central zone of cataractous lens epithelium was confirmed. Similar observations were made in the anterior capsule obtained during cataract surgery by Fu et al. (2016) [11] and Turan and Turan (2020) [56]. In addition, a significant reduction in the percentage of spheres from the central zone of the cataractous lens epithelium was identified in the present study. Functional loss of adult stem cells in diseased conditions, e.g., exhaustion of myogenic stem cells in dystrophic muscle, indicate the significance of adult tissue-resident stem cells in maintaining tissue homeostasis [57]. The absence of SOX2^+^ cells and the presence of autofluorescing vacuoles in the central zone of native tissue, as well as the reduction in the percentage of sphere forming ability of cultured lens epithelial cells in the cataractous lens, highlights the cellular changes in the central zone of cataract lens epithelium. Further studies are essential to explore the association of these changes with cataract pathogenesis.

## 5. Conclusions

In summary, this study identified four distinct zones in the human anterior lens epithelium and confirmed the location of the putative adult lens epithelial stem cells in the central zone and the functional loss of these stem cells in the cataractous condition.

## Figures and Tables

**Figure 2 cells-12-02727-f002:**
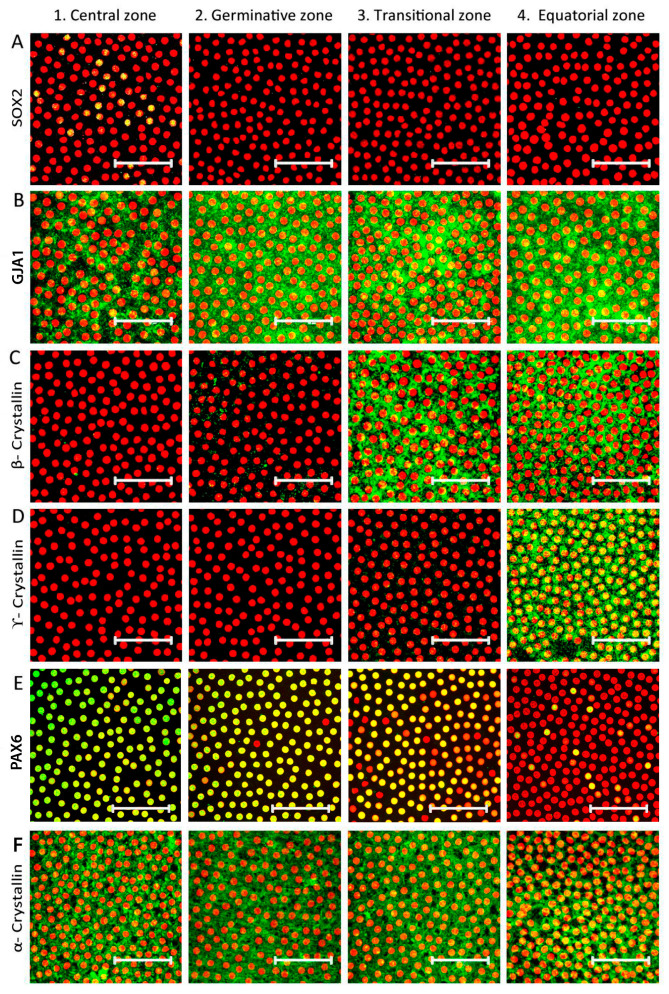
Identification of four zones based on specific markers in the human anterior lens epithelium. Representative confocal images of human anterior lens epithelial whole mounts immunostained for the markers (FITC-green) (**A**) SOX2, a stem cell marker, (**B**) GJA1, a differentiated cell marker, (**C**) β-crystallin, an early cell elongation marker, (**D**) γ-crystallin, late elongation marker, (**E**) PAX6, a neuroectodermal marker and (**F**) α-crystallin, counterstained with propidium iodide (PI-red). Expression of SOX2 was observed in a few cells (green + red = yellow) in the central zone, while cells in the three other zones were negative (**A**); the presence of a small proportion of cells negative for GJA1 membrane expression, β-crystallin and γ-crystallin defined the central zone, while GJA1^+^ β-crystallin^−^ γ-crystallin^−^ cells confirmed the germinative zone (**B**); the GJA1^+^ β-crystallin^+^ γ-crystallin^−^ cells defined the transitional zone (**C**); and the equatorial zone was characterized by late elongation marker γ-crystallin (**D**). The nuclear expression of neuroectodermal marker PAX6 was observed in almost all cells from central to transition zone and only in a small population in the equatorial zone (**E**).The lens epithelial cell marker α-crystallin was expressed throughout the epithelium (**F**). Scale bar represents 50 μm.

**Figure 3 cells-12-02727-f003:**
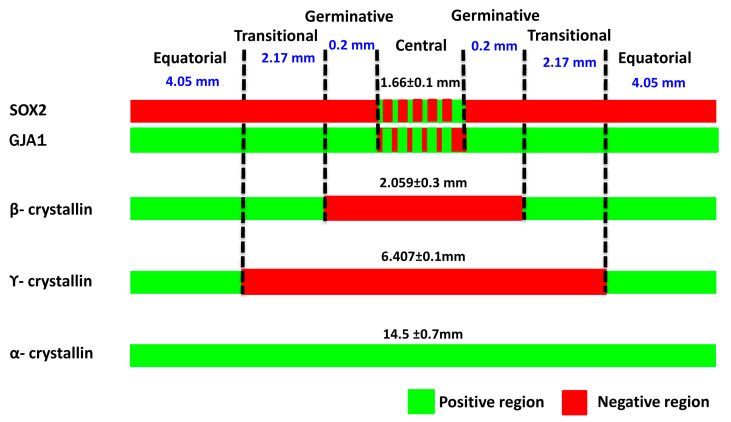
Characterization of four zones in the human anterior lens epithelium. Based on the expression of markers GJA1 and β-, γ- and α-crystallins in the lens epithelium, the length of the four zones was identified.

**Figure 4 cells-12-02727-f004:**
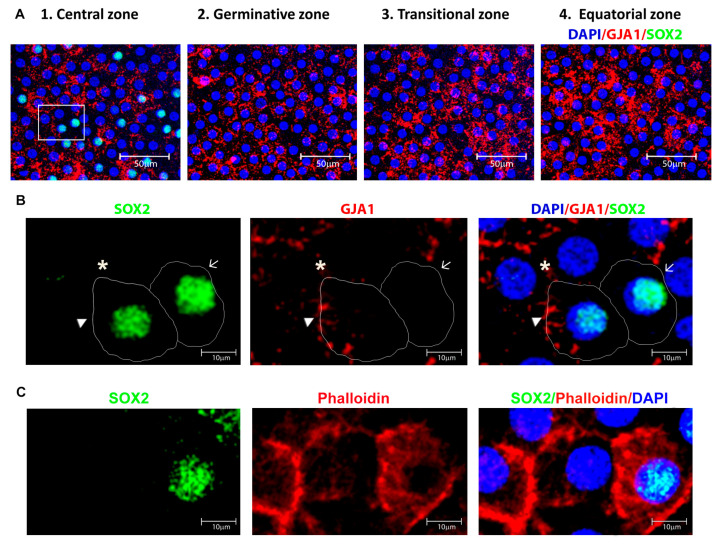
(**A**) Characterization of cells in the central zone of the human anterior lens epithelium. Representative confocal images of the whole mount of the human anterior lens epithelium immunostained for SOX2 (FITC-green) and GJA1 (Alexa 633-red), counterstained with 4,6-diamidino-2-phenylindole (DAPI-blue). The cells positive for SOX2 (green + blue = cyan) were present in the central zone. All of the epithelial cells in the germinative, transitional and equatorial zones were negative for SOX2 but positive for GJA1. (**B**) Higher magnification of the highlighted region (box in 4A) in the central zone indicated the presence of three populations of cells: SOX2^+^ GJA1^−^ cell (arrow); SOX2^+^ GJA1^+^ cell (arrow head) and SOX2^−^ GJA1^+^ cell (asterisk). The cells expressing SOX2 but negative for GJA1 were defined as putative stem cells, and such cells were restricted to the central zone. (**C**) Representative confocal imaging of the whole mount of the human anterior lens epithelium immunostained for SOX2 (FITC-green) and phalloidin (TRITC-red) highlighted the outline and structure of the lens epithelial cells.

**Figure 5 cells-12-02727-f005:**
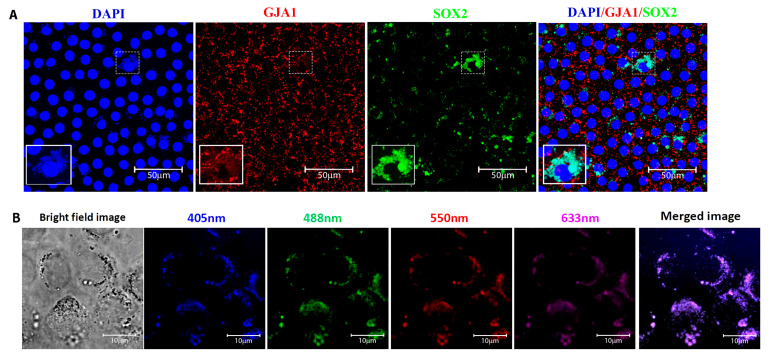
(**A**) Characterization of the cells in the central zone of cataractous human anterior lens epithelium. Representative confocal microscopic images of the whole mount of human cataractous lens epithelium immunostained for SOX2 (green) and GJA1 (red) counterstained with DAPI (blue). All of the epithelial cells were negative for SOX2 but positive for GJA1 (as evident from the red punctate expression in the inset of the overlay image). (**B**) The unstained cataractous donor lens whole mount revealed the presence of cytoplasmic vacuoles only in the central zone (bright field image). These vacuoles emitted autofluorescence (entire spectrum: UV to far red).

**Figure 6 cells-12-02727-f006:**
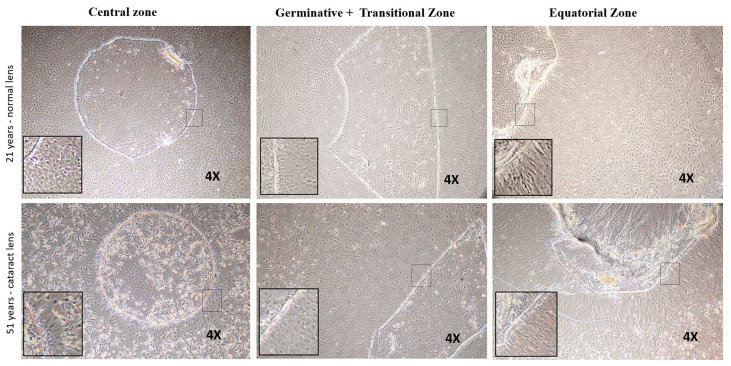
Representative phase contrast microscopic images of the explant cultured human lens epithelial cells from three zones of normal and cataract lens. In normal lens (**upper panel**): cells proliferated from the central zone were observed to be cuboidal; germinative and transitional zone showed a mixed population, and the equatorial zone revealed that the cells migrated from the explant were fibre-like cells. In cataract lens (**lower panel**): cells from central and germinative+transitional zones showed a mixed population while those from equatorial zone were fibre-like cells.

**Figure 7 cells-12-02727-f007:**
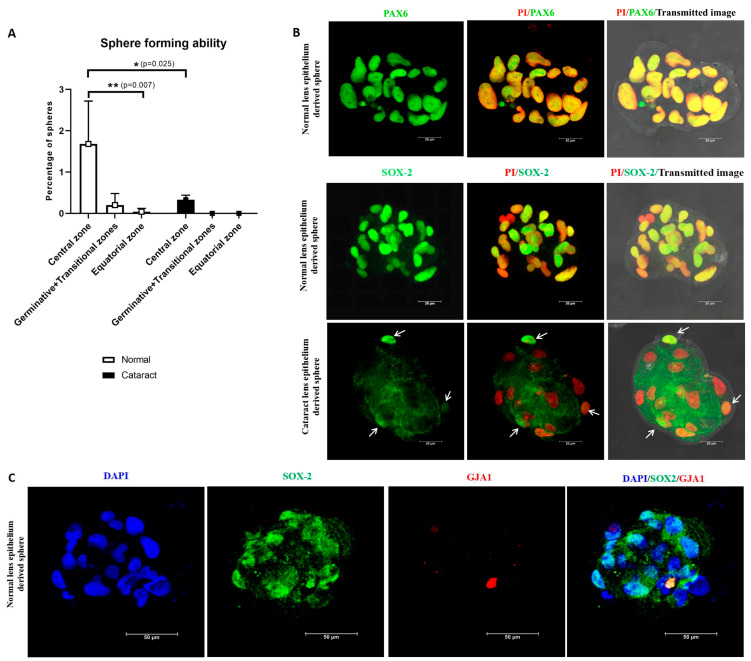
(**A**) The sphere forming ability of the cultured lens epithelial cells was analysed in both normal and cataract lens epithelium. The central zone alone had high sphere forming ability in normal lens epithelium, and a significant reduction was evident in cataract lens. * *p* = 0.025; ** *p* = 0.007. (**B**) Characterization of the human lens spheres derived from cultured central zone epithelial cells. Representative confocal microscopic images of lens spheres immunostained for PAX6/SOX2 (green) counterstained with propidium iodide (PI—red). The expression of PAX6 was evident in all cells in the sphere, and SOX2 was observed in most of the cells in the normal lens epithelium-derived spheres, while only a few cells in the spheres derived from the cataractous lens expressed SOX2 (arrows). Scale bar 20 μm. (**C**) Representative confocal microscopic images of a sphere derived from central zone double immunostained for SOX2 (green), GJA1 (red), counterstained with DAPI (blue). The SOX2 positive cells in the sphere were negative for GJA1.

**Table 2 cells-12-02727-t002:** Primary and secondary antibodies used in this study.

Antibody	Species and Type	Manufacturer, City/State, Country	Catalogue Number	Dilution Factor
SOX2	Rabbit monoclonal	Abcam, Waltham, MA, USA	ab92494	1:100
Connexin-43 (GJA1)	Mouse monoclonal	BD Biosciences, Franklin Lakes, NJ, USA	610062	1:200
PAX6	Rabbit polyclonal	Proteintech, Manchester, UK	12323-1-AP	1:250
α-crystallin	Mouse monoclonal	SantaCruz Biotechnology, Dallas, TX, USA	sc-28306	1:50
β-crystallin	Mouse monoclonal	SantaCruz Biotechnology, Dallas, TX, USA	sc-374496	1:50
γ-crystallin	Mouse monoclonal	SantaCruz Biotechnology, Dallas, TX, USA	sc-365256	1:50
Phalloidin-TRITC	Synthetic: Peptide Sequence	Sigma, Saint Louis, MO, USA	P1951	1:400
anti-mouse	Goat polyclonal	Agilent, Santa Clara, CA, USA	E0433	1:200
anti-rabbit	Mouse monoclonal	SantaCruz Biotechnology, Dallas, TX, USA	sc-2491	1:200

**Table 3 cells-12-02727-t003:** Distribution of cells expressing the stem cell marker SOX2 and differentiated epithelial cell marker GJA1 in the four zones of the whole mount of the human anterior lens epithelium. The putative stem cells, defined as SOX2^+^ GJA1^−^ cells, were present only in the central zone.

S. No	Zone	Percentage of Cells (Mean ± SEM) (*n* = 3 Donor Eyes)		
		SOX2^+^ GJA1^−^	SOX2^+^ GJA1^+^	SOX2^−^ GJA1^+^
1	Central zone	1.89 ± 0.84	7.27 ± 5.07	90.83 ± 5.9
2	Germinative zone	0	0	100
3	Transitional zone	0	0	100
4	Equatorial zone	0	0	100

**Table 4 cells-12-02727-t004:** Percentage of spheres (≥60 μm) was analysed in normal (*n* = 5) and cataractous (*n* = 3) lens epithelial explant cultured cells. Central zone had higher percentage of sphere forming ability in normal lens, and that was significantly reduced in cataractous lens.

Spheres Formed/10^3^ Cells (Spheres ≥ 60 μm)			
Age (Years)	Central Zone (% of Spheres)	Germinative + Transitional Zone (% of Spheres)	Equatorial Zone (% of Spheres)
Spheres established from normal lens epithelium			
18	1.3	0.6	0.2
20	1.1	0	0
21	3.1	0.4	0
41	2.4	0	0
54	0.5	0	0
Mean ± SD	1.68 ± 1.04	0.2 ± 0.28	0.04 ± 0.08
Spheres established from cataractous lens epithelium			
51	0.4	0	0
53	0.4	0	0
60	0.2	0	0
Mean ± SD	0.33 ± 0.11	0	0

## Data Availability

Data are contained within the article and Appendix A.

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
