# Peer review of "Towards the Identification and Characterization of Putative Adult Human Lens Epithelial Stem Cells"

_cells, 2023, doi:10.3390/cells12232727_

Round 1
Reviewer 1 Report
Comments and Suggestions for Authors
In this submission, the authors examine primary human lens cells, both in situ by whole mount immunofluorescence staining and after culture, to identify lens epithelial stem cells in normal and cataractous lenses. From their studies, they conclude that “The stem cells defined as SOX2+ Cx43- cells were identified to be located only in the central zone of human anterior lens epithelium and the significantly higher percentage of sphere formation from this zone confirmed the location.”
The precise definition of a stem cell is often contentious and can be a matter of semantics. These authors, as well as others, aim to define a specific subset of lens epithelial cells as bona fide stem cells. This study does not examine proliferation directly (e.g., label retention of BrdU), and attempts to identify subsets of lens epithelial cells by their pattern of expression of Sox2 (expressed in some types of stem cells), Cx43 (which encodes a gap junction protein), and alpha, beta, and gamma crystallins. The expression pattern of the aforementioned crystallins observed in adult human lenses in the study fits with what has previously been reported. The novel observation reported in this submission is “Thus, the lens epithelial stem cells identified as SOX2+ Cx-43- cells were confined to the central zone of human lens epithelium.” The data that support this conclusion are presented in Fig 4B, which shows Sox2 staining in two central lens epithelial cells claimed to be negative for Cx43. As a gap junction protein, Cx43 staining in lens epithelial cells has been well characterized as punctate staining at cell-cell interfaces (DeRosa et al., Exp Cell Res. 2009; White et al., IOVS, 2007). This is quite different from what looks like diffuse intracellular staining shown in Figs 2 and 4 that seems indistinguishable from staining for the cytoplasmic crystallins. In fact, the only staining that looks like canonical cell-cell interface gap junctional staining of Cx43 is at the plasma membrane of the two Sox2+ cells in Fig 4B claimed to be Cx43-negative. The data presented therefore do not support the conclusions drawn.
That said, the authors demonstration of Sox2 expression in different regions of adult human lens epithelium appears to be a novel contribution worthy of publication, provided the claim of a Sox2+, Cx43- stem cell population is either deleted or supported by additional data. The authors should also provide more data to support their claim that staining for Sox2 (without additional markers such as Abcg2 and examination of proliferative capacity) is a bona fide indicator of adult stem cells in human lens epithelium.
Additional comments:
It is debatable how well formation of spheroids from lens cells cultured in 20% fetal calf serum (an environment nonphysiological for the avascular lens) reflects stemness of lens cells. If, as is claimed, these highly Sox2+ spheroids are formed mainly from central epithelial “stem cells,” are they Cx43-, but Pax6+? The spheroid-forming culture medium contains 20 ng/ml FGF + heparin. Can the authors explain why the presence of such high levels of FGF does not induce their cultured lens cells to differentiate into lens fiber cells, as has been demonstrated in several in vivo and in vitro animal models?
Fig 5B shows that that whole mounts of cataractous donor lens contain cytoplasmic vacuoles in the central zone that emit autofluorescence throughout the entire UV to far red spectrum. The staining for Sox2 in these cells shown in Fig 5A is therefore deemed non-specific. Why, then is the staining for Cx43 (which to me also looks quite cytoplasmic, and is shown to co-localize with the Sox2 staining) considered specific?
Author Response
We thank you very much for taking the time to review this manuscript. Please find the detailed responses below.

Reviewer 2 Report
Comments and Suggestions for Authors
The manuscript by Saranya Pandi, Madhu Shekhar, Haripriya Aravind, Muthukkaruppan Veerappan and Gowri Priya Chidambaranathan “Identification and functional characterization of adult humanlens epithelial stem cells in normal and cataractous lens” attempts for the first to define adult human lens epithelial stem cells. This study is based on early findings from Hochedlinger lab [ref 26] from 2011 via genetic lineage tracing using Sox2-GFP reporter mice with in depth analyses of stomach, testis and other organs and tissues. Lens studies are limited to Fig. 2F showing Sox2 expression in the adult mouse lens epithelium. The authors need to expand their Introduction and include early studies on Sox2 and Sox1 in Kondoh lab, e.g. PMID: 9609835. Unfortunately, there are no data available on Sox2 downregulation of expression following strong upregulation of Sox1 in murine lenses. Arnold’s data show Sox2+ cells in epithelia > 3 weeks. Since forced expression of Sox2, together with Oct4, Klf4 and c-Myc, in somatic cells results in the dedifferentiation and transition to pluripotency it is necessary to further comment on these proteins and lens biology, namely roles of c-Myc and N-Myc. In the Introduction, additional paragraph is needed to explain in depths meridional rows, see studies from Bassnett lab (PMID: 28411123 and 25816743). As of now, Figure 2 shows a small subpopulation of SOX2+ cells within the central zone of lens epithelium and validates expression of individual crystallin classes. Figure 4 adds additional resolution of the signals, including phalloidin and DAPI. Figure 5 analyzes epithelial samples from cataract patients. Table 3 shows percentage of cells marked by SOX2 and Cx-43 (GJA1, see below). Finally, cells obtained from three zones were tested for their “sphere forming ability”. Taken together, this manuscript provides interesting though limited insights into a small subpopulation of SOX2+ cells within the “central zone” and is within the scope of the special issue of Cellsorganized by Dr. Lovicu.
Additional comments:
1) Title: Current title does not fit with the actual findings.
2) Introduction/Discussion: Lens regeneration may be presented with additional details and references, e.g. PMID: 16414361 and 26958831.
3) Connexin 43 has an official name GJA1. Cx-43 evokes mouse protein.
Author Response
We thank you very much for taking the time to review this manuscript. Please find the detailed responses below (document enclosed).

Round 2
Reviewer 1 Report
Comments and Suggestions for Authors
see attached PDF; needed to use a PDF since I had images in the review

Author Response
We thank the reviewer for their evaluation, and comments, which we have addressed (Document enclosed for your kind perusal).

Reviewer 2 Report
Comments and Suggestions for Authors
Both reviewers raised the same major issue, i.e. that SOX2+ human lens epithelial cells are insufficiently characterized in order to call them "lens stem cells". Thus, additional minor revisions are needed. In the Discussion, it is possible to formulate a testable hypothesis that some or all SOX2+ cells may represent "putative lens stem cells" and additional markers of these cells have to be discovered. In addition, mouse model has to be established involving lineage-tracing experiments. Finally, there is no any a priori reason that SOX2, due to its role in the inner cell mass/ES cells, has to be expressed in tissue-specific adult stem cells.
Author Response

(The authors gave the same response as above.)

Round 3
Reviewer 1 Report
Comments and Suggestions for Authors
The authors have substantially revised their manuscript in response to the reviewers’ concerns. Two key issues remain, which need to be addressed by changes to the text, and an addition to a figure.
1. In their response, the authors state that anti-Cx43 immunoreactivity near the nucleus is not on the plasma membrane (“But the single speckled positivity observed was near the nucleus and not in the plasma membrane.”) In order to make this statement, the cells would need to be double-labeled with a plasma membrane marker, which they are not. The authors must explicitly define what they consider Cx43 plasma membrane staining, and acknowledge that this is not verified by co-staining with a plasma membrane marker. In Fig 4B, the authors need to outline their assumed cell plasma membranes to justify labeling the cell with the arrow as “Cx43-“ and the cell marked with the arrowhead as “Cx43+.” The reader can judge for themselves if they agree with this assignment.
The sequential Z-stack images shown as supplementary information do not address this critical issue, and appear to show a slightly cropped version of the field shown in the maximum projection.
2. The title remains:
Adult human lens epithelial stem cells: Identification, characterization, and changes in cataractous condition
although the authors acknowledge that they have not identified lens stem cells. (“ As suggested by the reviewer, the discussion is now modified as given below formulating a hypothesis that SOX2+GJA- cells might represent “putative stem cells” and the need to identify stem cell specific markers..”)
The strongest title the evidence supports is
Towards the identification and characterization of putative adult human lens epithelial stem cells
3. Similarly, the authors’ statement in the abstract:
(line 26) Compared to other zones, a significant percentage of spheres was identified in the central zone (1.68±1.04%) confirming the location of adult lens epithelial stem cells. (bold mine)
Is at the very least an overinterpretation. Must be restated:
Compared to other zones, a significant percentage of spheres was identified in the central zone 25 (1.68±1.04%) consistent with the location of the putative adult lens epithelial stem cells.
4. Also:
Line 247: 3.2. Identification and Location of adult stem cells in human anterior lens epithelium
The adult stem cells in human lens epithelium were identified by the expression of the transcription factor SOX2.
Must be qualified with “proposed” or “putative” adult stem cells.
5. And:
Line 366: In this study, the adult stem cells in the human anterior lens epithelium were characterized based on the immunostaining for stem cell marker SOX2 along with differentiated cell marker GJA1 (plasma membrane expression) and by sphere forming assay in normal and cataractous lens.
Must be qualified with “proposed” or “putative” adult stem cells.
6. And:
Line 389: The adult tissue resident stem cells were identified by immunophenotyping using the transcription factor SOX2 as a marker. “
Must be qualified with “proposed” or “putative” adult tissue resident stem cells
Minor:
In 7C, it is claimed “The SOX2 positive cells in the sphere were negative for GJA1.” What Is their interpretation of the single Cx43-staining positive cell?
Line 397: Thus, the SOX genes SOX2 and SOX1 are essential for lens epithelium and fibre differentiation respectively.
What is the evidence they are essential, instead of merely present?
Author Response
On behalf of all the authors, I thank the reviewer for the valuable comments and suggestions.

Reviewer 2 Report
Comments and Suggestions for Authors
None.
Author Response
We thank the reviewer for accepting our manuscript.